# Three-Dimensional Cell Culture Micro-CT Visualization within Collagen Scaffolds in an Aqueous Environment

**DOI:** 10.3390/cells13151234

**Published:** 2024-07-23

**Authors:** Sergey Tkachev, Natalia Chepelova, Gevorg Galechyan, Boris Ershov, Danila Golub, Elena Popova, Artem Antoshin, Aliia Giliazova, Sergei Voloshin, Yuri Efremov, Elena Istranova, Peter Timashev

**Affiliations:** 1Institute for Regenerative Medicine, Sechenov University, Moscow 119991, Russia; 2Laboratory of Clinical Smart Nanotechnologies, Institute for Regenerative Medicine, Sechenov University, Moscow 119991, Russia; 3World-Class Research Center “Digital Biodesign and Personalized Healthcare”, Sechenov University, Moscow 119991, Russia

**Keywords:** micro-CT, confocal microscopy, spatial biology, bioinspired materials, collagen scaffolds, in vitro models, regenerative medicine

## Abstract

Among all of the materials used in tissue engineering in order to develop bioequivalents, collagen shows to be the most promising due to its superb biocompatibility and biodegradability, thus becoming one of the most widely used materials for scaffold production. However, current imaging techniques of the cells within collagen scaffolds have several limitations, which lead to an urgent need for novel methods of visualization. In this work, we have obtained groups of collagen scaffolds and selected the contrasting agents in order to study pores and patterns of cell growth in a non-disruptive manner via X-ray computed microtomography (micro-CT). After the comparison of multiple contrast agents, a 3% aqueous phosphotungstic acid solution in distilled water was identified as the most effective amongst the media, requiring 24 h of incubation. The differences in intensity values between collagen fibers, pores, and masses of cells allow for the accurate segmentation needed for further analysis. Moreover, the presented protocol allows visualization of porous collagen scaffolds under aqueous conditions, which is crucial for the multimodal study of the native structure of samples.

## 1. Introduction

Collagen is the most abundant fibrous protein in nature. It is widely present in the extracellular matrices of tissues such as connective tissue, tendon, skin, bone, and cartilage. Collagen provides mechanical support in tissues, and it gives essential cues to the surrounding cells, playing an important role in controlling cell adhesion, cell migration, differentiation, and tissue repair [1]. Therefore, due to its biological compatibility and nativity, collagen is the ideal biomaterial for us as a source for the production of tissue-engineered scaffolds or constructs [2]. Among others, collagen-based products are the most widely used in clinical practice, and they have been applied in a number of fields, such as dentistry, combustiology, neurology, urology, and cardiology [3], which makes this biomaterial highly relevant for investigation in the current study.

In the context of tissue engineering, scaffolds composed of native, full-length collagen fibers offer numerous attractive features. Advancements in fabrication techniques have enabled the creation of various collagen scaffold forms, including sponges, hydrogels, films, fibers, and meshes. Porous, three-dimensional (3D) collagen sponges have been utilized in various applications, such as wound dressings, tampons, 3D models of tissue-engineered constructs with diverse cell types, regenerative endodontic therapy, and as substitutes for bone cartilage [4,5].

A scaffold for tissue engineering should be a porous structure with highly interconnected pores that enable nutrient and oxygen diffusion, biological waste removal, and a 3D environment for cell assembly and differentiation. The pore, the size of which is in the range of 150–250 µm, is on average optimal for a cell distribution and smooth cell delivery onto the scaffold, while it may vary beyond these limits depending on the type of cells used in the experiment [6]. The determination of scaffold porosity is essential for the creation of an optimal artificial extracellular matrix that is suitable for various tissue types. Furthermore, a crucial aspect of fabricating tissue-engineered 3D constructs is the spatial visualization of cells within the scaffold, as a uniform cell distribution throughout is vital for the successful functioning of the artificial tissue.

To achieve the right design of collagen scaffolds and tissue-engineered constructs, as well as proving their biological compatibility and functionality, it is essential to correctly visualize their structure and changes during maturation and after implantation [7]. However, the existing imaging techniques for cells within collagen scaffolds have several limitations. The standard technique for visualization is serial sectioning followed by histological staining. Despite providing a good visualization of the cells, the resulting image is two-dimensional (2D), and the complex 3D architecture of the scaffold, as well as the true distribution of the cells, is irreversibly compromised during sample processing. Optical techniques, such as confocal laser scanning microscopy, multiphoton fluorescence microscopy, and light sheet microscopy, can be used to image cells (including live cell imaging) and their distribution within 3D constructs. To date, significant resolution improvement has been achieved, and the spatial resolution of confocal microscopy reaches 100–200 nm laterally and 350–500 nm axially [8,9,10,11]. However, these techniques do not provide a global view of the scaffold’s geometry and structure, e.g., the pore distribution, quantification of the closed and open (interconnected) porosity, or characterization of pore volume and shape. For instance, in confocal reflection microscopy, the fibers perpendicular to the imaging plane cannot be detected due to a blind spot, resulting in incomplete information in 3D images after reconstruction [12]. Additionally, the penetration depth of confocal microscopy is determined by the aperture of the objective and the diameter of the pinhole and limited by light absorption by the material of scaffolds due to their relatively enlarged thickness compared to the layers of cells [10]. Ultrasound imaging techniques enable the assessment of the compositional features of constructs, such as cell size and number, but the spatial resolution of 3D ultrasound-rendered images is limited [13].

The aforementioned limitations lead to an urgent need for novel visualization techniques that allow the imaging of the seeded cells as well as the 3D structure of the scaffold while preserving its native architecture. Given the importance of designing collagen-based products, micro-CT imaging, together with confocal laser scanning and light sheet microscopy, could be used as a method of multimodally examining constructs to evaluate the cellular distribution and structural features of the scaffolds.

X-ray computed microtomography (micro-CT) is a 3D, non-destructive imaging technique that enables the visualization of specimens with a minimum voxel size of 1–5 µm^3^ [14]. Over the past decade, significant improvements in micro-CT technology, along with the development of new sample processing protocols, have made feasible the high-resolution 3D imaging of non-mineralized biological objects, including tissues, 3D cellular cultures, and scaffolds, while preserving their native architecture [15,16,17,18,19]. Modern tomographic software, equipped with advanced methods of image processing, enables the comprehensive analysis of micro-CT datasets. This includes the visualization of internal structures, the determination of porosity, density analysis, the segmentation of specimen parts, complex geometric measurements, and a wide array of other applications [16,17,18,19,20,21,22,23]. Recent studies also demonstrate at least partial compatibility with conventional histological and histochemical techniques (depending on the contrasting agent), and, therefore, both imaging methods can be combined to obtain a multimodal image of the same specimen, where micro-CT and histological techniques complement each other [20,21,24].

Although micro-CT has proven to be a valuable technique for imaging bone structures, the visualization of soft tissue remains challenging due to its low inherent X-ray contrast. One way to enhance the contrast is to utilize contrast agents [25]. The introduction of radiopaque contrast agents (CAs) made high-resolution micro-CT a very important tool in biomedical imaging since these CAs bind to tissues of interest or perfuse through them, thus increasing their X-ray attenuation coefficient [26].

An interest in visualizing non-mineralized biological objects has led researchers to develop a variety of different staining protocols that can be suitable for diverse applications. For example, iodine-based contrast is universal but has low specificity, while there are agents that allow the staining of collagen (phosphotungstic acid, PTA [16,27]), neurons (Golgi–Cox method [28]), and even individual cells (contrasts using lead acetate [29,30] and eosin [31]). At the same time, contrast agents may require certain conditions, which influence the planning of the experiment and guide the experiment itself, prompting the choice of agent. For example, when studying a collagen scaffold colonized by cells, it is crucial to preserve its architecture, which is possible only in aqueous conditions for sustaining its hydrated native-like state. For this reason, the main candidate contrasting agent may be phosphotungstic acid, which is well-soluble in water and actively binds to positively charged collagen groups [32]. Furthermore, PTA staining does not significantly affect the structure of tissues and cells. Studies have shown that, following histological examination, samples did not reveal subsequent cellular destruction after the PTA dye exposure; however, this staining does not preserve cell viability [33,34,35]. Therefore, PTA staining is widely applied in the field of non-disruptive tissue investigation in developmental biology and the 3D investigation of soft tissue. There are protocols with PTA staining that allow the achievement of an enhanced contrast for CT examination without damaging the organ.

The possibility of visualizing a collagen scaffold in aqueous conditions was previously demonstrated in a study by Kwon et al. [16], in which scaffolds were contrasted with this agent and their porosity was subsequently analyzed (however, they used a 0.1% concentration of PTA solution). However, the scaffolds themselves were not populated, and thus the simultaneous 3D visualization of cells and collagen matrix in an aqueous environment has not yet been performed. In a work by Palmroth et al. [36], cells and collagen matrix were visualized using iron oxide nanoparticle staining, but the samples were dehydrated, which alters their architecture and pore and cell arrangement and thus does not adequately reflect their native structure and cell distribution. An additional objective is to segment the cells for the subsequent analysis of their volume and distribution inside the scaffold, specifically discrimination between live and dead cells at the moment of scaffold fixation. For this purpose, both basic contrasts such as phosphotungstic acid and specialized ones aimed at visualizing cellular structures (for example, contrasts based on nanoparticles, lanthanoides, and lead acetate) can be used. It is important to note that other specialized contrast agents (for example, lead acetate, which is nucleus-specific) interact weakly with collagen, and, thus, it becomes necessary in any case to use phosphotungstic acid for a high-quality visualization of the scaffold itself when the investigation is performed in a wet environment. However, currently, there are no studies that aim to determine the compatibility of two or more contrasts for this objective. Ghezzi et al. also demonstrated the possibility of visualizing cells inside collagen scaffolds using micro-CT after PTA treatment [37]. However, the authors did not use porous scaffolds and, in addition, used an ethanol solution of phosphotungstic acid, which could affect the pore structure due to the dehydration of the samples; hence, this protocol was not acceptable for our study. Nevertheless, this work also demonstrated the possibility of using PTA to simultaneously visualize the structure of collagen scaffolds and the cells inside them.

The objective of this study is to develop and initially validate a novel protocol for visualizing the spatial distribution of cells within collagen scaffolds after staining in 3% PTA under aqueous conditions using X-ray microtomography. Furthermore, we characterize scaffolds using porosity analysis with the segmentation of the pores. Additionally, we aim to investigate the potential for distinguishing between living and dead cells during the contrasting and segmentation processes from the collagen matrix based on intensity levels in the different zones of the specimen.

## 2. Materials and Methods

### 2.1. Collagen Sponge Matrix Preparation

The processes of obtaining and working with collagen scaffolds were conducted in an isolated environment under sterile conditions. The collagen sponge matrix was manufactured at the Center for Collagen Innovation within the Institute of Regenerative Medicine at Sechenov University and provided to us for experimental purposes. In order to obtain the collagen, the authors utilized animal-derived materials sourced from the tendons of large-horned cattle. To do this, the tendons were cleaned of excess tissue, cut into pieces with a thickness of 1 cm, and sequentially treated for 12 h in a 0.5 M NaCl solution. Subsequently, the mass was homogenized in a 0.83 M acetic acid solution. The resulting suspension was hydrolyzed with 0.24% pepsin for 2 days, after which 1 M NaOH was added to adjust the pH to 7.5, halting the hydrolysis process. The suspension was precipitated with a 12% NaCl solution, and the resulting precipitate was redissolved in 0.02 M acetic acid and then dialyzed. To obtain collagen porous matrices (sponges), the obtained solution was neutralized using 0.1 M NaOH until a pH of 7–7.5 was reached, and the resulting suspension was lyophilized at −40 °C for 2 days.

Subsequently, the collagen matrix was cut into cubes with sides measuring 1 cm. These cubes were placed in 15 mL test tubes filled with 70% ethyl alcohol for sterilization. The test tubes were then placed on a shaker and left in the refrigerator at +4 °C for 24 h. Afterward, the collagen matrices were removed from the alcohol and rinsed five times with 0.9% NaCl.

Following the alcohol rinse to confirm the absence of toxicity, an elution test, adapted following the ISO 10993 protocol [19], was conducted. To obtain collagen cube extracts, they were incubated in a cell culture medium at a volume of 1 mL per sample for 24 h at 37 °C. The 3T3 cell culture was passaged, with 5000 cells seeded in each well of a 96-well plate. After 24 h, the cells were treated with extract at a volume of 200 µL per well and left in the incubator at 37 °C for 24 h. The following day, extracts were collected, and alamarBlue reagent (Invitrogen, Waltham, MA, USA) was added according to the manufacturer’s instructions to assess the metabolic activity of the cells. Serial dilutions of sodium dodecyl sulfate (SDS) were used as the positive control. The graph representing the results of the collagen sponges’ indirect cytotoxicity test was plotted using Microsoft Excel 2020 (Microsoft, Redmond, Washington, DC, USA) software. Fluorescence intensity was measured using a Victor Nivo spectrofluorometer (PerkinElmer, Waltham, MA, USA) at an excitation wavelength of 530 nm and an emission wavelength of 590 nm.

### 2.2. Cell Experiments

After confirming the absence of cytotoxic effects, collagen sponges were seeded with the NIH 3T3 cell line at a density of 50,000 cells per sample (cubes of collagen sponge measuring 0.5 cm per side). NIH 3T3 culture medium was prepared based on Dulbecco’s Modified Eagle Medium F12 (DMEM/F12, 1:1, Biolot, Saint-Petersburg, Russia), gentamicin (50 μg/mL, PanEco, Moscow, Russia), and fetal calf serum FBS (10%, Thermo Fisher Scientific, Waltham, MA, USA). Cultivation was carried out in an incubator under standard conditions (37 °C, 5%CO_2_). After 3 days, to visualize the presence of cells in the samples, cryosections were prepared from a portion of the sponges for histological examination and staining of cell nuclei with 4′,6-diamidino-2-phenylindole (DAPI).

Briefly, collagen sponges were fixed in a freshly prepared 4% formaldehyde solution in phosphate-buffered saline (PBS) and stored at +4 °C overnight. Subsequently, the samples were transferred to a 30% sucrose solution and left overnight at +4 °C. The samples were then rinsed of sucrose, placed in optimal cutting temperature (OCT) compound (Servicebio, Wuhan, Hubei, China), and 20 μm thick cryosections were prepared using a cryostat (Thermo Scientific, model HM525 NX).

A portion of the sections was stained with hematoxylin and eosin. The samples were washed in PBS three times, then stained with hematoxylin for 5 min, followed by 70% ethanol and a 2-min eosin stain. Subsequently, the samples were rinsed with 95% ethanol and fixed. Images were obtained using a light-emitting diode microscope (LED) Leica DM1000 (Leica Microsystems, Wetzlar, Germany).

Another set of samples was rinsed three times in PBS and stained with the nuclear dye DAPI (Servicebio, Wuhan, Hubei, China) at a 1:500 dilution for 10 min. Evaluation of the distribution of cell nuclei was performed using a digital inverted microscope EVOS M5000 (Thermo Fisher Scientific, Waltham, MA, USA).

After confirming the presence of cells in the 3D collagen structures, samples were stained using the Live/Dead method to detect live and dead cells within the samples. Live cells were stained with 0.5 mg/mL Calcein-AM (Calcein-AM, Sigma-Aldrich, Darmstadt, Germany), and dead cells were stained with 1.5 µM propidium iodide (Thermo Fisher Scientific, Waltham, MA, USA). Stained sponges were incubated at 37 °C with 5% CO_2_ for 30 min. Subsequently, the samples were rinsed three times with DMEM/F12 medium (1:1, Biolot, Saint-Petersburg, Russia) and analyzed using an Olympus FV3000 confocal microscope (Olympus, Tokyo, Japan). Subsequently, the samples were prepared and stained for micro-CT visualization.

An isotonic lanthanoid-containing solution BioREE-A (Glaucon, Moscow, Russia) was used in our work for lanthanoid treatment. A detailed protocol for lanthanoid treatment of cells is described in the work of Subbot et al. [38].

### 2.3. Staining Technique

All specimens were divided into dedicated groups based on different contrast staining techniques and presence of living or dead cells. We used phosphotungstic acid alone and in combination with lead acetate and lanthanoids. Lead acetate or lanthanoid treatment was performed to test the possibility of specific staining of cells within the scaffolds.

Based on the protocols used, contrasting was performed as follows:

#### 2.3.1. Phosphotungstic Acid

Fixed specimens in 10% formalin with PBS were washed after 24 h with distilled water and, after that, placed in 3% phosphotungstic acid dissolved in distilled water for 24 h and kept on the rotary shaker at room temperature. After staining, samples were washed and stored in distilled water at 5 °C.

#### 2.3.2. Lead Acetate and Phosphotungstic Acid

Fixed specimens in 10% formalin with PBS were washed after 24 h with distilled water and, after that, placed in 10% lead acetate for 18 h. The lead acetate technique was based on the work by Metscher et al. [17]. After that, samples were washed with distilled water and placed in 3% phosphotungstic acid, as described above.

#### 2.3.3. Lanthanoides and Phosphotungstic Acid

Fixed specimens pretreated with lanthanoids were placed in 3% phosphotungstic acid, as described above.

### 2.4. Micro-CT Image Acquisition

A plastic tube filled with distilled water containing the contrasted sample was placed on the sample holder in a SkyScan 1276 micro-CT (Bruker, Kontich, Belgium) and scanned using the following parameters: for the Al1 mm filter, the source voltage was 70 kV, and for the AlCu filter, the source voltage was 90 kV. Source current was 200 μA for both filters. Sample was scanned with 3–6 μm voxel resolution. For each filter, the rotation was set to 180° or 360° around the vertical axis of the sample, with two middle frames for each 0.2° angle step. The source current and pixel size were varied with the aim of identifying the best scanning parameters.

### 2.5. Photography of the Unstained Scaffolds

Photography of the unstained scaffolds was performed using Olympus SZ-51 stereomicroscope (Olympus Olympus Co. Ltd., Tokyo, Japan) with mounted ADF PRO08 digital camera ADF Optics, Hangzhou, China) and ADF Image Capture software, version 4.1.21522.20221011(ADF Optics, Hangzhou, China).

### 2.6. Volume Rendering and Processing

After scanning, the data were reconstructed using Bruker NRecon software, version 1.6.9. During reconstruction, the ring artifact reduction value was set to 20%, and the beam hardening correction value to 30%.

Many advanced approaches have been used to perform visualization of reconstructed micro-CT images. Two-dimensional and three-dimensional visualization, as well as intensity analysis, were performed using the open-source software 3D Slicer, version 5.2.1. Segmentation of pores, fibers, and cell masses was performed using Bruker CTAn as well as image binarization. The resulting segmented images were visualized using Bruker CTVox and 3D Slicer. Porosity analysis was performed using Bruker CTAn software, version 1.17.7.2. Initially, a small rectangle within the boundary of the scaffold was created. Then, the ROI was exported as a series of 8-bit images in bmp format. Then, a series of the filters was applied in CTAn to the resulting image stack for specimen segmentation and preprocessing. Finally, 3D analysis was performed, which yielded detailed information about porosity in the sheet format and the 3D model for further 3D visualization in CTVox. Segmentation of the cellular masses was performed in a similar way and was based on high intensity values of the cells. 3D models of the cellular mass were made using a 3D modeling algorithm in CTAn and were further visualized in the CTVol program. For the simultaneous segmentation of fibers and cell masses, Bruker CTan was used to export binarized images from ROIs with intensity values specific to fibers and cell masses, respectively. The resulting images were then loaded into Bruker CTVox where they were volume-rendered and pseudo-colored.

### 2.7. Intensity Level Analysis and Statistics

Intensity level analysis was performed using 3D Slicer instrument “Probe”. We measured the intensity in 400 random points of the 16-bit images for each type of object: water, fibers, pores, and cells. After that, the resulting comma-separated values (CSV) table was analyzed and plotted using Python packages numpy, matplotlib, and seaborn.

## 3. Results

### 3.1. Evaluation of Collagen Sponges’ Indirect Cytotoxicity

The principle of the alamarBlue assay is based on the reduction of resazurin to resorufin under the action of mitochondrial enzymes in living cells. As a result, the blue color of the solution changes to pink. According to the results of comparing the metabolic activity of cells in the experimental group with the control, no toxicity was detected in the experimental group (Figure 1).

### 3.2. Two-Dimensional Visualization of Collagen Scaffolds Seeded with Cells

During staining cryosections using the nuclear stain DAPI, we succeeded in visualizing cell clusters in different regions of the collagen matrix. However, histological staining did not yield a similar pattern. We attribute this to the fact that hematoxylin and eosin staining is more aggressive. The use of ethanol severely dehydrated cellular structures and delicate collagen fibers, which could have facilitated cell detachment and washout during the staining process [39,40,41]. As a result, we were only able to visualize individual cells in the sections, whereas DAPI staining clearly revealed large cell aggregates (Figure 2 and Figure 3).

Visualization of live and dead cells using the Live/Dead method allowed us to obtain a more comprehensive view and confirm the presence of live cells in the samples. Since full-sized samples were used for visualization on a confocal microscope without mechanical manipulation, we were able to visualize a larger number of cells. Despite the fact that, nowadays, it is possible to create 3D confocal reconstructions [10,11], unfortunately, in this case, the spatial arrangement of these cells remained unclear. We suggest that the thickness of the scaffold allows visualizing cells only within a specific “slice” in a single plane (Figure 4).

### 3.3. Contrast Enhancement of the Scaffolds Is a Prerequisite for Analysis of Their Native Internal Structure in Aqueous Conditions

Although the dehydrated scaffold can be visualized without contrast agents, its internal structure and pore size are significantly altered compared to the hydrated scaffold. Uncontrasted scaffold under aqueous conditions is almost completely transmissive to X-rays, and, therefore, it is necessary to use water-soluble contrast agents to visualize it (Figure 5A,B).

### 3.4. Lead Acetate in Combination with PTA Is not Suitable for Contrasting Cells and Scaffolds

At first, we assumed that adding lead acetate, which selectively binds to cell nuclei, would allow better imaging than simple staining with PTA for the further visualization of both cells and collagen fibers. We also hypothesized that using only lead acetate would allow the visualization of isolated cells without collagen fibers. This assumption was based on previous studies demonstrating that lead acetate or its compounds enable the visualization of individual cells [29,30]. However, in our case, the combination of lead acetate and phosphotungstic acid created a significant number of artifacts. Thus, we conclude that this is not appropriate for contrasting cells in collagen scaffolds (Figure 5C,D).

### 3.5. The Combination of Lanthanoids and Phosphotungstic Acid Does Not Allow for Adequate Separation of Cells and Collagen Fibers

Contrary to our initial expectations, the lanthanoids stained not only the cells but also the collagen fibers. Although we noted clusters of cells when stained with lanthanoids alone, we also observed the partial staining of fibers, which created artifacts and, therefore, did not allow the mapping of cells within the scaffold. When further contrasting with PTA, we did not notice any significant differences from the usual contrasting with phosphotungstic acid (Figure 5E,F). There was also no difference in contrast accumulation between live and dead cells (at the moment of fixation). Thus, we can conclude that such a combination is not suitable for the analysis of cells inside collagen scaffolds.

### 3.6. Phosphotungstic Acid Treatment Enables Characterization of Pore Network and Simultaneous Visualization of Cell Masses and Collagen Scaffolds

Contrasting with a 3% aqueous PTA solution allows the visualization of collagen sponges under aqueous conditions while maintaining the original configuration of fiber and pore arrangement (Figure 6B,C). A good signal-to-noise ratio made it possible to segment fibers and pores for further analysis, which is important for the quality control of scaffolds and the study of their internal structure (Figure 6D–F).

In the samples we studied, we found that cell masses, scaffold fibers, water, and pores have different intensity values (Figure 7A–C). For instance, Figure 7B (unpopulated scaffold, a control sample) and Figure 7C (scaffolds seeded with cells) show images (without any post-processing except basic reconstruction) with color-coding based on intensity values in the identical modes of scanning and reconstruction. It allows the visualization of significant differences in X-ray densities between fibers, water, and pores. The intensity values of pores are slightly higher than those of the water surrounding the sample, and we assume that this is due to the beam-hardening effect that occurs when the X-ray beam passes through the sample (Figure 7A). It is worth mentioning that scaffolds can be labeled for their spatial orientation.

The cellular masses differed significantly in terms of X-ray density (Figure 7A,C,D) from the surrounding collagen fibers, and, therefore, it became possible to distinguish them and to carry out segmentation. The cellular masses appeared as single clusters or large conglomerates, deforming the surrounding collagen fibers during their growth. Often, as is shown in 3D reconstructions with an applied maximum intensity projection volume rendering technique, the growing cellular masses bulged through the existing pores in the scaffold, which gave their clusters the shape of interwoven “spaghetti” (Figure 7D and Figure 8A,B). It is worth mentioning that phosphotungstic acid does not allow the distinguishing of individual cells within clusters due to the chemical process of contrasting, where PTA preferentially binds with collagen and positively charged groups of proteins [35,42] and not with nuclear material such as nucleus-specific contrasts [29,30,31].

Similarly, the visualization and segmentation of collagen fibers permit a detailed examination of the scaffold’s internal architecture and the determination of its characteristics, such as porosity, pore connectivity, and fiber thickness. Segmented cell masses, or scaffold structures, such as specific regions of collagen fibers, can be converted into a mesh (Figure 8A) or exported for advanced analyses, such as evaluating cell growth patterns and their interaction with the collagen fibers (Figure 8B).

Using micro-CT image processing software for the visualization and segmentation of cells not only facilitates the qualitative assessment of their distribution within the scaffold but also enables the quantification of their volume (Table 1). We determined that the internal porosity of the scaffolds averaged 58%, and the pore size ranged from 150 to 250 µm. The volume of segmented cells ranged from small clusters of 0.2 mm^3^ volume to large conglomerates with volumes up to 8.6 mm^3^. Intensity values for cells were, on average, greater than those for collagen fibers by 147%, which allowed them to be segmented and distinguished from collagen fibers.

### 3.7. Phosphotungstic Acid Does Not Allow Distinguishing of Live or Dead Cells

We found no significant differences in morphology and intensity values between dead and live cells at the time of fixation. Both live and dead cells (at the moment of fixation) accumulated contrast equally, and, thus, processing specimens with PTA could not be used as a Live/Dead staining.

### 3.8. Artifacts and Pseudopositive Results

Perhaps the most significant disadvantage is that very dense, interwoven fibers, or clusters of fibers mimicking cell masses due to heterogeneous microparticles of collagen raw material, can form inside collagen scaffolds during production (Figure 9). To alleviate the issue described above, we have identified some criteria that will allow us to distinguish cell clusters from high-density fibers. Firstly, cells grow in free pore spaces within the scaffold, whereas dense fibers generally follow the normal geometry of fibers. During their growth within free pore spaces, cells can acquire peculiar geometries resembling intertwined tubes forming a large conglomerate (Figure 7C,D, Figure 8A and Figure 9A,B). Secondly, during its growth, the colony of cells can deform the surrounding fibers by shifting them with its growing mass, which is expressed in the appearance of deformations along the course of fibers (Figure 9A,B). Thus, although it is not exact, it is possible to identify an area of interest for further analysis. To segment cell masses, it would also be useful to use algorithms that allow the cleaning of the 3D model of unnecessary parts that could be fibers with high density (Figure 8A). An example of such an algorithm is Despeckle in CTAn, which has different modes of sweeping the image [43].

## 4. Discussion

In this study, we have performed the imaging of porous collagen scaffolds colonized by cells using X-ray microtomography. We compared several contrast agents and determined that the most adequate contrast protocol was based on the use of a 3% aqueous PTA solution. This contrast method allowed the simultaneous visualization of collagen fibers, pores, and cell masses, with the possibility of subsequent analysis. As the scaffolds were contrasted under aqueous conditions, their native internal architecture remained close to that before contrasting, providing an accurate understanding of their internal structure, porosity, and cell distribution.

In the samples we studied, we found that cell masses, scaffold fibers, and pores have different intensity values. The intensity values of pores are slightly higher than those of the water surrounding the sample, and we assume that this is due to the beam-hardening effect that occurs when the X-ray beam passes through the sample. Due to differences in intensity values, it is possible to separate pores, scaffold fibers, and cell masses from each other using different segmentation algorithms (e.g., Riddler–Calvard method). In turn, this allows not only the creation of sufficiently accurate 3D models but also the performance of various mathematical calculations. For example, in this study, we determined the porosity of scaffolds and measured the volume of segmented cell masses. Similarly, it is also possible to perform pore connectivity analysis [44], determine fiber directions [45], and calculate wall thicknesses (e.g., cell mass thicknesses) [19], which has been shown in previous studies of various objects using micro-CT imaging techniques. An interesting potential application of this imaging approach is to investigate cell migration within scaffolds and the relationship between the mechanical properties of the extracellular matrix, the cellular phenotype, and the mechanosensing machinery [46,47]. This may be particularly relevant for examining malignant cells in 3D cultures for their ability to migrate and invade depending on their phenotype and mechanical properties of the environment [48,49,50].

However, the presented methodology has its own drawbacks that need to be considered in future studies. First of all, due to the inherent inability of PTA contrasting to distinguish individual cells, in our work, we were able to observe only the conglomerates that accumulated contrast. On the other hand, it should also be considered that there are more powerful machines whose resolution is higher than the micro-CT scanner we used. These include nano-CT machines [29] and synchrotron phase-contrast tomography facilities, on which individual cells have been visualized in other studies [51]. Also, PTA staining cannot serve as a Live/Dead test, although we anticipate that cells that have died some time before contrast will have a lower intensity, and, thus, for example, necrotic cores within cell colonies can be visualized. Since we were unable to visualize live and dead cells separately, we propose the use of our methodology as a complement to confocal microscopy and the Live/Dead staining, as demonstrated in our study. Confocal microscopy would enable the clear identification of live cells within a 3D sample, while micro-CT could assist in determining their spatial distribution within the scaffold.

Although this method has some imperfections related to the physicochemical properties of the contrast agent, it is possible to implement it in cell biology and bioengineering studies if certain guidelines are followed. For example, collagen scaffold fibers can be easily segmented from the surrounding water in order to study their morphology as well as the morphology of the pore space. However, in the case of cell masses, care should be taken when analyzing images to avoid false positives. Since cells, dense collagen fibers, and collagen conglomerates can give a high-intensity signal, the main criteria for analysis are the morphological properties of the growing colony of cells, such as growth between collagen fibers within the pore space and the deformation of fibers along the growth path.

Potentially, this method can also be applied with other imaging techniques to obtain a multimodal image of the collagen scaffold populated with cells. Several visualization methods were used in our study (histologic staining, cell nuclei staining with DAPI, and Live/Dead staining), but, in fact, any other method from cell biology can be applied to create a more complete picture. Collagen scaffolds after PTA contrast were studied by scanning electron microscopy, and the possibility of histologic examination in micro-CT-defined areas has been demonstrated in several studies, including those where phosphotungstic acid was used as contrast.

Although, in our study, the combination of lanthanoids and PTA did not effectively separate individual cells, we hypothesize that it may be possible to develop a selective agent that would make this possible. As a theoretical suggestion for future research, we propose to develop an imaging protocol based on the different signatures characteristic of PTA and a hypothetical contrast agent, and then perform segmentation based on the difference in these signatures. Moreover, in a similar way, a whole palette of agents could also be created that selectively bind to different cells and ensure their further precise segmentation, which could be essential for research in the field of cell biology. Based on previous research, it is also worth noting that machines with the best possible resolution, such as nano-CT and high-resolution synchrotron phase contrast tomography, should be used. Although previous work by Bhartiya and colleagues showed the imaging of cancer cells on an electrospun scaffold without the use of a contrast agent [52], the imaging was performed using synchrotron radiation. At the moment, this can only be done using synchrotron radiation sources, the number of which in the world is not large [53]. Because of high demand, the waiting time for the experiment can be long, while the laboratory micro-CT scanner allows several images to be taken in a fairly short time: before the cells are populated with scaffolds, to assess their structure and pore network, and after, to assess the distribution of cell masses.

We assume that micro-CT with contrast enhancement using PTA could be applied for the spatial visualization of scaffolds that consist of materials in which PTA could selectively bind with positively charged molecules, such as fibrin, elastin, and silk fibroin [16,35]. However, since one should test the compatibility of PTA staining in an aqueous environment with other scaffold materials, further investigations are required.

The protocol presented in our study can be used with a laboratory tomography scanner, which has recently become quite common equipment in biomedical research. Moreover, the rich experience gained over the last 10 years in analyzing biological objects can be used for the more accurate analysis of 3D cell cultures, including advanced image segmentation techniques using machine learning technologies [54]. Thus, a new simple but powerful tool for assessing the spatial distribution of cells within bioengineered scaffolds is emerging in cell biology, expanding the range of applications of micro-CT in biomedical research.

## 5. Conclusions

In this study, porous collagen scaffolds populated by cells were imaged using X-ray microtomography in an aqueous environment. The optimal contrast protocol was identified as 24 h incubation in a 3% aqueous PTA solution. Differences in intensity values between collagen fibers, pores, and cell masses allow accurate segmentation for further analysis while preserving the scaffold’s native architecture. The presented protocol allows the visualization of porous collagen scaffolds under aqueous conditions, which is crucial for studying the native structure of samples. In our study, we have demonstrated the feasibility of potential sample analysis techniques including internal structure visualization, pore and fiber segmentation, and cell mass segmentation followed by 3D model construction. Therefore, we suggest that such techniques could be useful in the fields of cell biology and regenerative medicine for the detailed investigation of 3D cell cultures and scaffolds.

## Figures and Tables

**Figure 1 cells-13-01234-f001:**
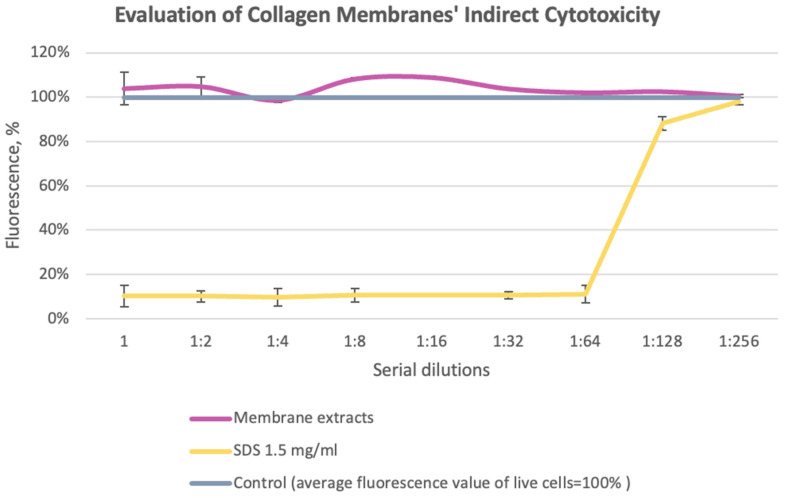
Comparison of metabolic activity of cells: experimental group (collagen sponge extract); control group (cells treated with sodium dodecyl sulfate (SDS)).

**Figure 2 cells-13-01234-f002:**
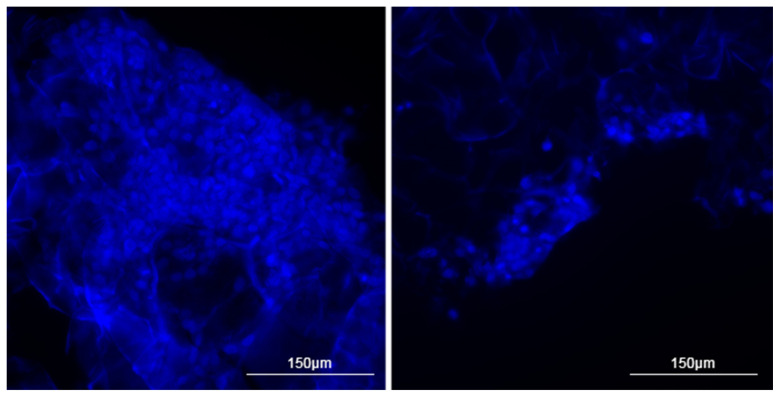
Staining of cryosections of collagen sponges using DAPI nuclear dye. Clusters of cell nuclei in the thickness of collagen fibers are visualized. Collagen fibers have a strong autofluorescence, so cell nuclei can be differentiated from fibers by their rounded shape. Magnification power is ×20.

**Figure 3 cells-13-01234-f003:**
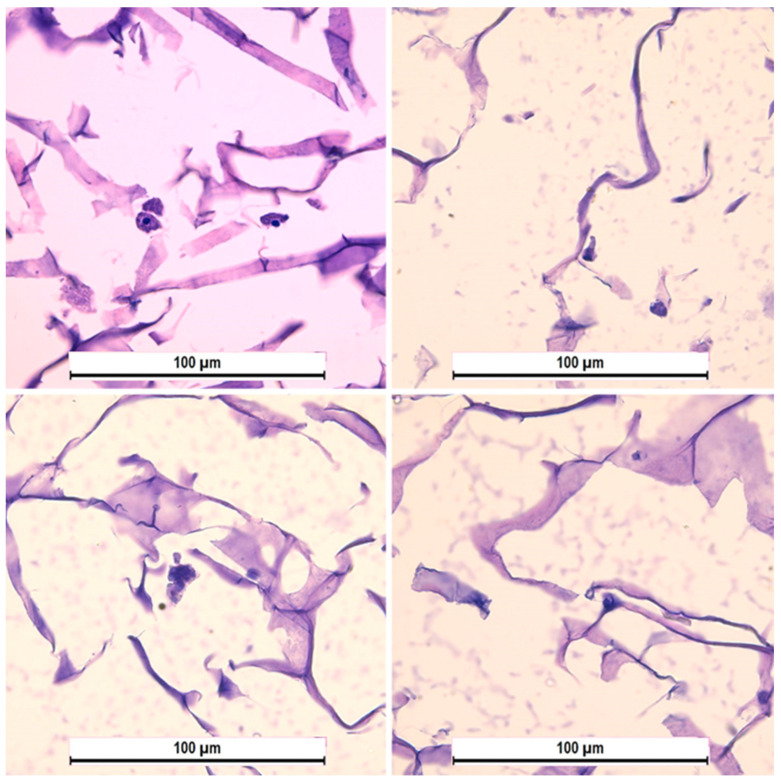
Hematoxylin and eosin staining of cryosections of collagen sponges. Single cells and damaged collagen fibers are visible. Magnification power is ×40.

**Figure 4 cells-13-01234-f004:**
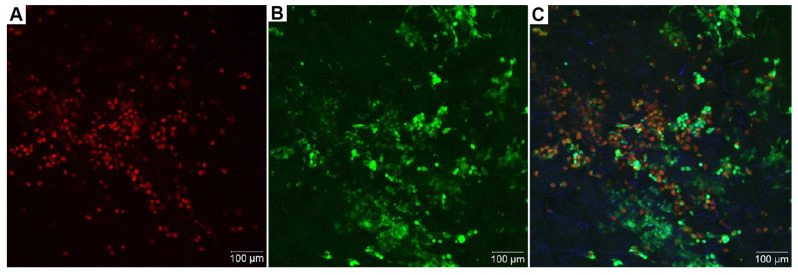
Live/Dead staining of collagen sponges and visualization on a confocal microscope. (**A**) Dead cells are stained red (PI); (**B**) Live cells are stained green (Calcein-AM); (**C**) Merged (**A**,**B**) image. Live cells are stained green (Calcein-AM); dead cells are stained red (PI); magnification power is ×20.

**Figure 5 cells-13-01234-f005:**
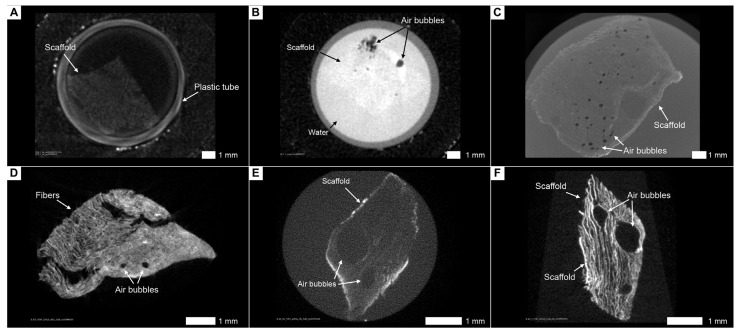
Visualization approaches that have proven ineffective. (**A**) Visualization of a dry, uncontrasted scaffold in air. (**B**) Visualization of an uncontrasted scaffold in water. (**C**) Lead acetate staining before PTA. (**D**) Lead acetate staining after PTA. (**E**) Lanthanoid staining before PTA. (**F**) Lanthanoid staining after PTA.

**Figure 6 cells-13-01234-f006:**
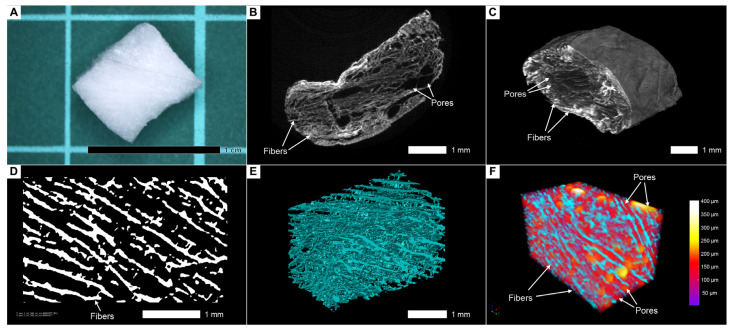
Visualization of the empty (unpopulated) collagen scaffolds before and after staining in a 3% PTA solution in water and different approaches for the image analysis. (**A**) Photography of the unstained collagen scaffold. (**B**) Two-dimensional micro-CT image of the unpopulated stained scaffold. (**C**) Volume-rendering model of the stained scaffold. (**D**) Segmentation of the fibers using thresholding and binarization. (**E**) Volume-rendering model of the segmented fibers. (**F**) Porosity analysis of the scaffold with the segmentation of the pores, color coding with based on the pore size. Segmented fibers labeled in blue. Differences in intensity values allow specialized software (such as Bruker CTAn, v. 1.17.7.2) to segment objects and pores. In this case, we overlaid the obtained images to visualize spatial distribution of pores between collagen fibers.

**Figure 7 cells-13-01234-f007:**
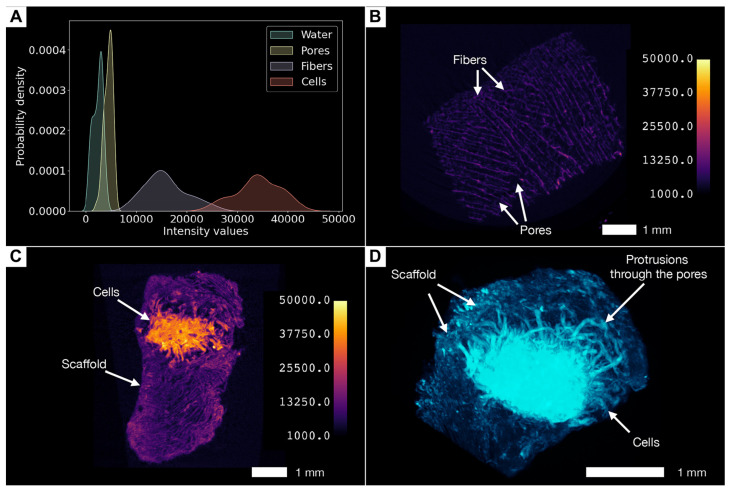
Visualization and analysis of the cells in the collagen scaffolds. (**A**) Differences in intensity values enable distinguishing between fibers, pores, and cell masses. (**B**) 2D micro-CT image of the empty collagen sponge (control) with color-coding based on intensity values. (**C**) 2D micro-CT image of the collagen scaffold populated with the 3T3 cells with color-coding based on intensity values. Note the high accumulation of the phosphotungstic acid by cells. (**D**) Maximum intensity projection of the same specimen with the appearance of large conglomerate of the cells, 3D micro-CT image.

**Figure 8 cells-13-01234-f008:**
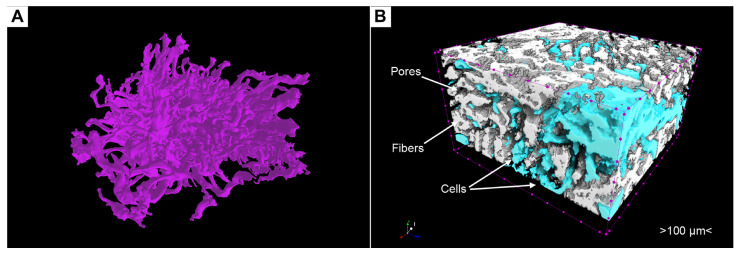
Advanced segmentation approaches. (**A**) 3D model (mesh) of the cellular colony based on the segmentation. (**B**) Visualization of the distribution of the cells inside the region of interest. Both cellular masses and collagen fibers were segmented and binarized based on intensity level. Such an approach allowed visualization of the spread of cell masses between the collagen fibers of the scaffold. This spreading pattern allows us to distinguish false positive signal from dense fibers because, in the case of cells, they spread within the pore network between the fibers.

**Figure 9 cells-13-01234-f009:**
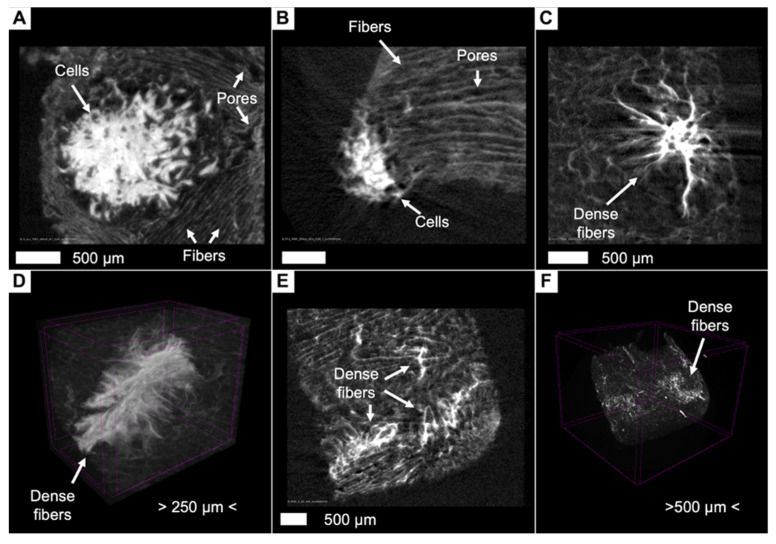
Artifacts and pseudo-positive results. (**A**,**B**). Cellular masses. Note significant deformations of the fibers in A due to growth of cellular colony. Images (**C**–**F**) represent pseudo-positive results due to dense collagen fibers mimicking cell masses in the 2D and 3D images with the maximum intensity projection modes.

**Table 1 cells-13-01234-t001:** Characterization of the scaffolds’ porosity and the volume of cellular clusters.

Parameter	Value
Average internal porosity, %	58
Min pore size, µm	150
Max pore size, µm	250
Min volume of segmented cells, mm^3^	0.2
Max volume of segmented cells, mm^3^	8.6

## Data Availability

The processed data required to reproduce these findings are available to download from URL https://doi.org/10.5281/zenodo.10177428 (accessed on 19 July 2024).

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
