# Peer review of "Three-Dimensional Cell Culture Micro-CT Visualization within Collagen Scaffolds in an Aqueous Environment"

_cells, 2024, doi:10.3390/cells13151234_

Round 1

Reviewer 1 Report

Comments and Suggestions for Authors

The article is interesting and clearly highlights the advantages of the method. However, there is a small lack of knowledge regarding the alternative techniques available and some gaps in the materials and methods section. In particular, what is presented in the material and methods section must be discussed in the article otherwise, why insert it there. Conversely, what is presented in terms of results deserves to be detailed in the material and methods section (e.g. cell culture conditions). I believe that all the shortcomings of this article can be easily corrected.

The author presents a method allowing the analysis of porous materials, such as a collagen scaffold, by X-ray microtomography in an aqueous medium. This method makes it possible to highlight the fibers and pores of the scaffold but also makes it possible to visualize the distribution of cells. Although several techniques already allow the analysis of cellular distribution in a matrix, few methods allow the structure of the scaffold to be simultaneously evaluated. As the author points out, micro-CT is indeed a formidable tool that tissue engineering laboratories should equip themselves with, but for this, new protocols to optimize imaging are necessary. Although I find this article interesting, I suggest the author make some corrections to improve the content.

1.      In the introduction, I disagree with the statement that it is not possible to scan more than 200 um with confocal or multiphoton microscopy. Many highly effective optical clarification techniques have been developed over the past 20 years. They allow to significantly reduce light scattering. I suggest the author review literature on this topic.

2.      In the materials and methods section, mention is made of a metabolic activity test carried out using the alamar blue reagent. Although the test seems very relevant to me here, no results are presented or discussed in connection with this test. It would be necessary for the author to fill this gap or remove the test from the method section.

3.      Still in the materials and methods section, I suggest adding information on cell culture conditions after adding cells on the collagen scaffold. Do not limiting the protocol on for producing the collagen sponge.

4.      On line 226, it says that the samples are placed in lead acetate for 18 hours but I don't understand if there are two distinct conditions: 2% or 10% solutions or if we rather mean that a first treatment at 2% is followed by a second at 10%, each of them for 18 hours.

5.      In the results section, Figure 1 shows blurred images. An image taken on a 20 um slice should not be this blurry unless out of focus. If possible, I suggest choosing a better images.

6.      In the second paragraph of section 3.1, it is false to claim that it is not possible to visualize the spatial arrangement of cells using confocal microscopy. See my comment at point 1.

7.      In Figure 3, at B, a Hoechst marking appears. I don't understand the point of this marking in this montage. Hoechst is very effective on fixed tissues because it then marks all the cells (all of which are dead). On the other hand, when it comes to living cells, Hoechst is not a good marker. As can be seen in the image, very few cells are labeled unlike the labeling with calcein and propidium iodide. Perhaps the author could explain to us the reason for his choice to use the Hoechst in this montage and what information we can draw from it. This is not covered in the article.

8.      In the caption of Figure 4, the description of photo A is the same as B and yet the two images are very different. Review the description of each. There must be a mistake.

9.      Figures 4, 6 and 7: It would have been interesting to see the controls without cells. This would have been particularly relevant in Figures 6 and 7 in order to discriminate the artifacts described by the author from the positive signal coming from the cells. Furthermore, knowing that a cell measures approximately 10-15 microns, greater magnification would have made it possible to better see the organization of the cells in the scaffold.

10.   It would have been easy to add a table presenting the results presented at the end of section 3.3. It should be emphasized that the method allows quantification.

Comments on the Quality of English Language

Minor editing of English language required.

Author Response

For research article “Three-Dimensional Cell Culture Micro-CT Visualization within Collagen Scaffolds in an Aqueous Environment”

Response to Reviewer 1 Comments

  1. Summary

We are grateful for your review and consideration for publication with Cells as well as the opportunity to respond to the reviewer comments for our manuscript. We appreciate the effort invested by the reviewers and the editorial team. Please find the detailed responses below and the corresponding corrections highlighted in the resubmitted files.

  1. Point-by-point response to Comments and Suggestions for Authors

General Comments: The article is interesting and clearly highlights the advantages of the method. However, there is a small lack of knowledge regarding the alternative techniques available and some gaps in the materials and methods section. In particular, what is presented in the material and methods section must be discussed in the article otherwise, why insert it there. Conversely, what is presented in terms of results deserves to be detailed in the material and methods section (e.g. cell culture conditions). I believe that all the shortcomings of this article can be easily corrected.

The author presents a method allowing the analysis of porous materials, such as a collagen scaffold, by X-ray microtomography in an aqueous medium. This method makes it possible to highlight the fibers and pores of the scaffold but also makes it possible to visualize the distribution of cells. Although several techniques already allow the analysis of cellular distribution in a matrix, few methods allow the structure of the scaffold to be simultaneously evaluated. As the author points out, micro-CT is indeed a formidable tool that tissue engineering laboratories should equip themselves with, but for this, new protocols to optimize imaging are necessary. Although I find this article interesting, I suggest the author make some corrections to improve the content.

Authors’ Response: We thank the reviewer for positive and encouraging feedback about our manuscript. We completely agree with aforementioned suggestions, and, according to the reviewer’s comments, we have added necessary clarifications in the manuscript. We have reviewed literature an extended the Introduction section. We have added necessary information about cell culture conditions in the Materials and methods, chapter “2.2 Cell experiments section, and evaluation of collagen sponges”. Also we have added results of the indirect cytotoxicity test in Results section, chapter “3.1 Evaluation of Collagen Sponges' Indirect Cytotoxicity”.

For the reader’s convenience, we have updated figures and described them in detail over the course of the text.  We have added new figures that more clearly show that cell masses, scaffold fibers, water and pores have different intensity values (Figure 7. A, B, C). We have obtained these images without any post-processing except basic reconstruction. Figure 7. B, C show images with color-coding based on intensity values in the identical modes of scanning and reconstruction. It allows visualizing significant differences of X-ray densities between fibers, water and pores.

Furthermore, we have added a separate paragraph “3.8 Artifacts and pseudopositive results” to clarify criteria and process of segmentation for distinguishing dense fibers and colonies of cells.

Comment 1: In the introduction, I disagree with the statement that it is not possible to scan more than 200 um with confocal or multiphoton microscopy. Many highly effective optical clarification techniques have been developed over the past 20 years. They allow to significantly reduce light scattering. I suggest the author review literature on this topic.

Response 1:  We thank the reviewer for pointing out the lack of knowledge in the Introduction section, and we apologize for our negligence. In a line with the reviewer’s comment, we have changed the original text (the Introduction section, paragraphs 4, 5, lines 65-84):  

[Optical techniques, such as a confocal laser scanning microscopy, multiphoton fluorescence microscopy, and the light-sheet microscopy, can be used to image cells inside 3D structures (including live cell imaging), but they only reach depths up to 200 μm [8,9] and do not provide a global view of the scaffold geometry. Additionally, light scattering is a significant problem for relatively thick scaffolds that reaches up 5mm [10]. Ultrasound imaging techniques provides assessing the compositional features of constructs, including cells size and number, but the spatial resolution of 3D ultrasound rendered images is limited [11]. The aforementioned limitations lead to an urgent need for novel visualization techniques, which allow an imaging of the seeded cells as well as three-dimensional structure of the scaffold while preserving its native architecture.]

to the following text:

         [Optical techniques, such as a confocal laser scanning microscopy, multiphoton fluorescence microscopy, and the light-sheet microscopy, can be used to image cells (including live cell imaging) and their distribution within 3D constructs. To date, significant resolution improvement has been achieved, and the spatial resolution of confocal microscopy reaches up 100-200 nm laterally and 350-500 nm axially [8–11]. However, these techniques do not provide a global view of the scaffold geometry and structure, e.g. the pore distribution, quantification of the closed and open (interconnected) porosity, characterization of pore volume and shape. For instance, in confocal reflection microscopy, the fibers perpendicular to the imaging plane cannot be detected due to a blind spot, resulting in incomplete information in 3D images after reconstruction [12]. Additionally, the penetration depth of confocal microscopy is determined by the aperture of the objective and the diameter of the pinhole, and limited by light absorption by the material of scaffolds due to relatively enlarged thickness compared to layers of cells [10]. Ultrasound imaging techniques provides assessing the compositional features of constructs, cells size and number, but the spatial resolution of 3D ultrasound rendered images is limited [13].

         The aforementioned limitations lead to an urgent need for novel visualization techniques, which allow an imaging of the seeded cells as well as 3D structure of the scaffold while preserving its native architecture. Given the importance of design collagen-based products, micro-CT imaging could be used as a method of multimodal examination of constructs together with confocal laser scanning and light-sheet microscopy to evaluate cellular distribution and structural features of the scaffolds.].

Comment 2: In the materials and methods section, mention is made of a metabolic activity test carried out using the alamar blue reagent. Although the test seems very relevant to me here, no results are presented or discussed in connection with this test. It would be necessary for the author to fill this gap or remove the test from the method section.

Response 2: Thank you for pointing out the need to include this information in the Results section, chapter “3.1 Evaluation of Collagen Sponges' Indirect Cytotoxicity”. To reflect reviewer’s suggestion, we have added a plot represented the metabolic activity test results (lines 295-298).

Comment 3: Still in the materials and methods section, I suggest adding information on cell culture conditions after adding cells on the collagen scaffold. Do not limiting the protocol on for producing the collagen sponge.

Response 3: Acknowledging the gap in the Materials and methods section, we have added information about the conditions for culturing cells on collagen scaffolds (paragraph 1, lines 196-198): [NIH 3T3 culture medium was prepared based on DMEM/F12 (1:1, Biolot, Russia), gentamicin (50 μg/mL, PanEco, Russia), fetal calf serum FBS (10%, Thermo Fisher, USA). Cultivation was carried out in an incubator under standard conditions (37°C, 5%CO2).]

Comment 4: On line 226, it says that the samples are placed in lead acetate for 18 hours but I don't understand if there are two distinct conditions: 2% or 10% solutions or if we rather mean that a first treatment at 2% is followed by a second at 10%, each of them for 18 hours.

Response 4:  We are sorry for our mistake and thank the reviewer for this pointing out. The samples were placed in 10% lead acetate for 18 hours once. Therefore, we have changed the sentence [Fixed specimens in 10% formalin with PBS were washed after 24 hours with distilled water and after that placed in 10% lead acetate for 18 hours.]

to the following text:

 [Fixed specimens in 10% formalin with PBS were washed after 24 hours with distilled water and after that placed in 10% lead acetate for 18 hours.] (lines 242-243).

Comment 5: In the results section, Figure 1 shows blurred images. An image taken on a 20 um slice should not be this blurry unless out of focus. If possible, I suggest choosing better images.

Response 5: Unfortunately, collagen has strong autofluorescence, due to this the cells do not look very clear. We left only two images where the outlines of cell nuclei look the brightest (Figure 2).

Comment 6: In the second paragraph of section 3.1, it is false to claim that it is not possible to visualize the spatial arrangement of cells using confocal microscopy. See my comment at point 1.

Response 6We agree with the reviewer’s suggestion. We have corrected the following sentence [However, the spatial arrangement of these cells remains unclear because confocal microscopy allows visualization only within a specific "slice" in a single plane (Figure 3).]

to

[Despite the fact that nowadays it is possible to create 3D confocal reconstructions [10,11], unfortunately, in that case the spatial arrangement of these cells remains unclear. We suggest that the thickness of the scaffold allows visualizing cells only within a specific "slice" in a single plane (Figure 4).] (lines 323-326).

Comment 7: In Figure 3, at B, a Hoechst marking appears. I don't understand the point of this marking in this montage. Hoechst is very effective on fixed tissues because it then marks all the cells (all of which are dead). On the other hand, when it comes to living cells, Hoechst is not a good marker. As can be seen in the image, very few cells are labeled unlike the labeling with calcein and propidium iodide. Perhaps the author could explain to us the reason for his choice to use the Hoechst in this montage and what information we can draw from it. This is not covered in the article.

Response 7: Based on the reviewed literature, DAPI is a more toxic dye compared to Hoechst (Analyzing Cell Death by Nuclear Staining with Hoechst 33342, Cold Spring Harb Protoc; 2016; doi.org/10.1101/pdb.prot087205). Therefore, Hoechst is more suitable for visualizing nuclei in live cells. Since we performed Live/dead staining, we chose Hoechst. However, we agree that cell nuclei were understained in our image. As we have mentioned in the Response 5, collagen has strong autofluorescence, therefore, the cells do not look very clear, and these phenomena has made difficult to detect nuclei in the 3D collagen sponge. We left only two images where the outlines of cell nuclei look the brightest. Therefore, we decided to keep images only in two channels (green for live cells and red for dead cells).

Comment 8: In the caption of Figure 4, the description of photo A is the same as B and yet the two images are very different. Review the description of each. There must be a mistake.

Response 8:  We are sorry for our typo and thank the reviewer for this pointing out. We have corrected figure caption and changed text [B. Visualization of a dry uncontrasted scaffold in air.] to [B. Visualization of an uncontrasted scaffold in water.] (lines 335-338).

Comment 9: Figures 4, 6 and 7: It would have been interesting to see the controls without cells. This would have been particularly relevant in Figures 6 and 7 in order to discriminate the artifacts described by the author from the positive signal coming from the cells. Furthermore, knowing that a cell measures approximately 10-15 microns, greater magnification would have made it possible to better see the organization of the cells in the scaffold.

Response 9:  In a line with the reviewer’s comment, we added Figure 7. B (unpopulated scaffold), C (scaffolds with seeded cells) with color-coding based on intensity values in the identical modes of scanning and reconstruction (lines 397-404). It allows visualizing significant differences of X-ray densities between fibers, water and pores. We have obtained these images without any post-processing except basic reconstruction.

Additionally, we have repeated the intensity level analysis, so we have measured the intensity in 400 random points of the 16-bit images (instead of 8-bit images in the previous version of the manuscript) for each type of object: water, fibers, pores and cells. Therefore, the meanings of probability densities have increased, and the ranges of intensity values have expanded on the new plot, Figure 7. A (lines 397-404).

To distinguish dense fibers of collagen and colonies of cells, we applied advanced segmentation approaches, which we have described in the updated Figure 8 (lines 412-418). Both cellular masses and collagen fibers were segmented and binarized based on intensity level. Such an approach allowed visualization of the spread of cell masses between the collagen fibers of the scaffold. This spreading pattern allows us to distinguish false positive signal from dense fibers because in the case of cells, they spread within the pore network between the fibers.

Additionally, in a line with the reviewer’s comment, we have added a separate paragraph “3.8 Artifacts and pseudopositive results” to clarify criteria and process of segmentation for distinguishing dense fibers and colonies of cells (lines 435-450):

[3.8 Artifacts and pseudopositive results

Perhaps the most significant disadvantage is that very dense interwoven fibers or clusters of fibers mimicking cell masses due to heterogeneous microparticles of collagen raw material can form inside collagen scaffolds during production (Figure 9). To alleviate the issue described above, we have identified some criteria that will allow us to distinguish cell clusters from high-density fibers. Firstly, cells grow in free porous spaces within the scaffold, whereas dense fibers generally follow the normal geometry of fibers. During the growth within the free pore spaces, cells can acquire peculiar geometries resembling intertwined tubes forming a large conglomerate (Figure 7. C, D; Figure 8. A; Figure 9. A, B). Secondly, during its growth, the colony of cells can deform the surrounding fibers by shifting them with its growing mass, which is expressed in the appearance of deformations along the course of fibers (Figure 9. A, B). Thus, although it is not exact, it is possible to identify an area of interest for further analysis. To segment cell masses, it would also be useful to use algorithms that allow cleaning the 3D model from unnecessary parts that could be fibers with high density (Figure 8. A). An example of such an algorithm is Despeckle in CTAn with different modes of sweeping the image [43].]

Also we highlighted that greater magnification could not make it possible to better see the organization of the cells in the scaffold due to non-selective binding of PTA with collagen and positively-charged groups of proteins (lines 394-396):

[It is worth mentioning that phosphotungstic acid does not allow to distinguish individual cells within clusters, due to the chemical process of contrasting where PTA preferentially binds with collagen and positively-charged groups of proteins [34,41], and not with the nuclear material like nucleus-specific contrasts [28–30].]

Comment 10: It would have been easy to add a table presenting the results presented at the end of section 3.3. It should be emphasized that the method allows quantification.

Response 10:  We thank the reviewer for this pointing out, so we have added the Table 1 in the end of section “3.6 Phosphotungstic acid treatment enables characterization of pore network and simultaneous visualization of cell masses and collagen scaffolds” (line 427).

  1. Response to Comments on the Quality of English Language

Point 1: Minor editing of English language required.

Response 1: We apologize for the typos and mistakes of our manuscript. We polished the language to improve readability of the manuscript. These changes did not influence the content and framework of the paper. Thus, we did not list or mark the changes in the revised paper. We really hope that the language level has been improved.

Reviewer 2 Report

Comments and Suggestions for Authors

The authors presented a novel imaging method that exploits X-ray to visualize collagen scaffolds and cells preserving the scaffold morphology. The topic is really interesting, however more information on the methods need to be provided.

Major

·       Paragraph 2.2: please provide more detail on how the sponges were seeded.

·       Paragraph 2.6: in the supplementary information the authors should provide one (or multiple) raw image in which the difference in pixel intensity is clearly showed to the reader. Moreover, the intensity ranges for each element (water, fibers, pores, and cells) need to be clearly reported in the manuscript.

·       Data from the cytotoxic experiments were described at row 168 are not reported in the results section. They should be added. If the authors prefer to not add them in the main text, they can add them in the supplementary information.

·       The difference among figure 4A and 4B is not clear in the label.

·       Figure 5F: more detail about how the authors obtain this figure need to be included in the manuscript.

·       Figure 7: it is not clear how the authors differentiate between dense fiber and cells.

·       The authors should discuss if their methodology could be used also for other materials.

Minor

·       The authors should specify in the abstract and introduction if they are proposing a visualization method that can be used with live cells and if their method is disruptive/non disruptive.

·       Please be sure that all the acronyms are defined

·       Line 276-278: please add references.

Comments on the Quality of English Language

none

Author Response

For research article “Three-Dimensional Cell Culture Micro-CT Visualization within Collagen Scaffolds in an Aqueous Environment”

Response to Reviewer 2 Comments

  1. Summary

We are grateful for your review and consideration for publication with Cells as well as the opportunity to respond to the reviewer comments for our manuscript. We appreciate the time and effort invested by the reviewers and the editorial team. Please find the detailed responses below and the corresponding corrections highlighted in the resubmitted files.

  1. Point-by-point response to Comments and Suggestions for Authors

General Comments: The authors presented a novel imaging method that exploits X-ray to visualize collagen scaffolds and cells preserving the scaffold morphology. The topic is really interesting, however more information on the methods need to be provided.

Authors’ Response: We thank the reviewer for positive and encouraging feedback about our manuscript. We have added necessary clarifications and corrections in the manuscript.

Comment 1: Paragraph 2.2: please provide more detail on how the sponges were seeded.

Response 1: In a line with the reviewer’s comment, we have added the following information about seeding of cells (lines 196-198): [NIH 3T3 culture medium was prepared based on DMEM/F12 (1:1, Biolot, Russia), gentamicin (50 μg/mL, PanEco, Russia), fetal calf serum FBS (10%, Thermo Fisher, USA). Cultivation was carried out in an incubator under standard conditions (37°C, 5%CO2).].

Comment 2: Paragraph 2.6: in the supplementary information the authors should provide one (or multiple) raw image in which the difference in pixel intensity is clearly showed to the reader. Moreover, the intensity ranges for each element (water, fibers, pores, and cells) need to be clearly reported in the manuscript.

Response 2: In a line with the reviewer’s comment, we added Figure 7. B, C with color-coding based on intensity values in the identical modes of scanning and reconstruction (lines 397-404). It allows visualizing significant differences of X-ray densities between fibers, water and pores. We have obtained these images without any post-processing except basic reconstruction. Additionally, we have repeated the intensity level analysis, so we have measured the intensity in 400 random points of the 16-bit images (instead of 8-bit images in the previous version of the manuscript) for each type of object: water, fibers, pores and cells. Therefore, the meanings of probability densities have increased, and the ranges of intensity values have expanded in the new plot, Figure 7. A (lines 397-404).  

Comment 3:  Data from the cytotoxic experiments were described at row 168 are not reported in the results section. They should be added. If the authors prefer to not add them in the main text, they can add them in the supplementary information.

Response 3: In a line with the reviewer’s comment, we have added data from the cytotoxic experiments in chapter “3.1 Evaluation of Collagen Sponges' Indirect Cytotoxicity”, Figure 1 “Comparison of metabolic activity of cells: experimental group (collagen sponge extract); control group (cells treated with SDS)”, lines (291-298).  

Comment 4: The difference among figure 4A and 4B is not clear in the label.

Response 4: We are sorry for our typo and thank the reviewer for this pointing out. We have corrected figure caption and changed text in Figure 5.B [B. Visualization of a dry uncontrasted scaffold in air.] to [B. Visualization of an uncontrasted scaffold in water.] (lines 335-338).

Comment 5: Figure 5F: more detail about how the authors obtain this figure need to be included in the manuscript.

Response 5: We agree with this comment, so we have added the explanation in the Figure 6.F caption (lines 372-375): [F. Porosity analysis of the scaffold with the segmentation of the pores, color coding with based on the pore size. Segmented fibers labeled in blue color. Differences in intensity values allow specialized software (such as Bruker CTAn) to segment objects and pores. In this case, we overlaid the obtained images to visualize spatial distribution of pores between collagen fibers.].

Comment 6:  Figure 7: it is not clear how the authors differentiate between dense fiber and cells.

Response 6: To distinguish dense fibers of collagen and colonies of cells, we applied advanced segmentation approaches, which we have described in the Figure 8 (line 412). Both cellular masses and collagen fibers were segmented and binarized based on intensity level. Such an approach allowed visualization of the spread of cell masses between the collagen fibers of the scaffold. This spreading pattern allows us to distinguish false positive signal from dense fibers because in the case of cells, they spread within the pore network between the fibers.

Additionally, in a line with the reviewer’s comment, we have added a separate paragraph “3.8 Artifacts and pseudopositive results” to clarify criteria and process of segmentation for distinguishing dense fibers and colonies of cells (lines 434-451):

[3.8 Artifacts and pseudopositive results

Perhaps the most significant disadvantage is that very dense interwoven fibers or clusters of fibers mimicking cell masses due to heterogeneous microparticles of collagen raw material can form inside collagen scaffolds during production (Figure 9). To alleviate the issue described above, we have identified some criteria that will allow us to distinguish cell clusters from high-density fibers. Firstly, cells grow in free porous spaces within the scaffold, whereas dense fibers generally follow the normal geometry of fibers. During the growth within the free pore spaces, cells can acquire peculiar geometries resembling intertwined tubes forming a large conglomerate (Figure 7. C, D; Figure 8. A; Figure 9. A, B). Secondly, during its growth, the colony of cells can deform the surrounding fibers by shifting them with its growing mass, which is expressed in the appearance of deformations along the course of fibers (Figure 9. A, B). Thus, although it is not exact, it is possible to identify an area of interest for further analysis. To segment cell masses, it would also be useful to use algorithms that allow cleaning the 3D model from unnecessary parts that could be fibers with high density (Figure 8. A). An example of such an algorithm is Despeckle in CTAn with different modes of sweeping the image [43].]

Comment 7:  The authors should discuss if their methodology could be used also for other materials.

Response 7: In a line with the reviewer’s comment, we have added in the Discussion section, paragraph 7, lines 534-538, the following text:

[We assume that micro-CT with contrast enhancement using PTA could be applied for spatial visualization of scaffolds that consist of materials, in which PTA could selectively bind with positive charged molecules, such as fibrin, elastin, and silk fibroin [14,33]. However, since one should test compatibility of PTA staining in aqueous environment with other scaffold materials, the further investigations are required.] 

Comment 8: The authors should specify in the abstract and introduction if they are proposing a visualization method that can be used with live cells and if their method is disruptive/non disruptive.

Response 8: In a line with the reviewer’s comment, we have clarified these details in the abstract (line 18) and in the Introduction section that this visualization technique is non-disruptive for scaffolds and cells or tissues, according to mentioned in the Introduction about features of phosphotungustic acid staining (paragraph 7, lines 112-118): [Furthermore, PTA staining does not significantly affect the structure of tissues and cells. Studies have shown that following histological examination of samples did not reveal subsequent cellular destruction after the PTA dye exposure [30–32]. Therefore, PTA staining is widely applied in the field of non-destructive tissues investigation in developmental biology and three-dimensional investigation of the soft tissues. There are protocols with PTA staining, which allow achieving an enhanced contrast for CT examination without damaging the organ.]

Furthermore, we tested the possibility of revealing the viability of cells at the moment before fixation, and our observations have revealed that both live and dead cells accumulated contrast equally and thus processing specimens with PTA could not be used as a live-dead staining (section Results, chapter “3.7 Phosphotungstic acid does not allow distinguishing of live or dead cells”, lines 430-433).

Comment 9: Please be sure that all the acronyms are defined

Response 9: Thank you for your comment, we apologize for our negligence. We have checked the entire manuscript to add all definitions of acronyms.

Comment 10: Line 276-278: please add references.

Response 10:  In a line with the reviewer’s comment, we added references in order to prove statement about dehydration and destroying of collagen fibers due to the use of ethanol during hematoxylin and eosin staining. This effect of ethanol on collagen fibers was described in Turunen et al. (doi.org/10.1016/j.jsb.2017.07.009), Oosterlaken et al. (doi.org/10.1002/jemt.23915), and in Dey et al. (doi.org/10.1007/978-981-19-6616-3_10) works, which we have mentioned in the following text (section Results, chapter “3.2 Two-dimensional visualization of collagen scaffolds seeded with cells”, lines 323-326): [The use of ethanol severely dehydrated cellular structures and delicate collagen fibers, which could have facilitated cell detachment and washout during the staining process [38-40].].

  1. Response to Comments on the Quality of English Language

Point 1: None.

Response 1: We thank the reviewer for the positive feedback and constructive comments to our manuscript.